# Shortwave Infrared InGaAs Detectors On-Chip Integrated with Subwavelength Polarization Gratings

**DOI:** 10.3390/nano13182512

**Published:** 2023-09-07

**Authors:** Huijuan Huang, Yizhen Yu, Xue Li, Duo Sun, Guixue Zhang, Tao Li, Xiumei Shao, Bo Yang

**Affiliations:** 1State Key Laboratories of Transducer Technology, Shanghai Institute of Technical Physics, Chinese Academy of Sciences, Shanghai 200083, China; huanghuijuan21@mails.ucas.ac.cn (H.H.); yuyizhen@mail.sitp.ac.cn (Y.Y.); sunduo@mail.sitp.ac.cn (D.S.); 15216703892@163.com (G.Z.); litao@masil.sitp.ac.cn (T.L.); shaoxm@mail.sitp.ac.cn (X.S.); 2Key Laboratory of Infrared Imaging Materials and Detectors, Shanghai Institute of Technical Physics, Chinese Academy of Sciences, Shanghai 200083, China; 3University of Chinese Academy of Sciences, Beijing100049, China

**Keywords:** shortwave infrared, InGaAs, polarization detection, subwavelength metal grating

## Abstract

Shortwave infrared polarization imaging can increase the contrast of the target to the background to improve the detection system’s recognition ability. The division of focal plane polarization indium gallium arsenide (InGaAs) focal plane array (FPA) detector is the ideal choice due to the advantages of compact structure, real-time imaging, and high stability. However, because of the mismatch between nanostructures and photosensitive pixels as well as the crosstalk among the different polarization directions, the currently reported extinction ratio (ER) of superpixel-polarization-integrated detectors cannot meet the needs of high-quality imaging. In this paper, a 1024 × 4 InGaAs FPA detector on-chip integrated with a linear polarization grating (LPG) was realized and tested. The detector displayed good performance throughout the 0.9–1.7 um band, and the ERs at 1064 nm, 1310 nm and 1550 nm reached up to 22:1, 29:1 and 46:1, respectively. For the crosstalk investigation, the optical simulation of the grating-integrated InGaAs pixel was carried out, and the limitation of the ER was calculated. The result showed that the scattering of incident light in the InP substrate led to the crosstalk. Moreover, the deviation of the actual grating morphology from the designed structure caused a further reduction in the ER.

## 1. Introduction

Besides the spectral and intensity information of the reflected and radiated light used in conventional imaging technology, the polarization characteristics of the reflected light of objects can also be used for detection, which depends on the inherent properties of their surface, such as structural characteristics [1], material characteristics [2], roughness [3], etc. The additional information including the polarization degree and the polarization angle can be obtained through infrared polarization imaging technology, which can increase the contrast of the target with the background as well as the signal-to-noise ratio [4]. With the advantages of strong penetration, long detection distance, good concealment and strong night detection ability, the infrared polarization imaging technology can effectively improve the detection system’s recognition ability and accuracy of the target [5] and meet the needs of remote sensing [6,7,8], target detection [9,10], biomedicine [11,12,13], materials science [14,15,16], face recognition [17,18] and so on.

The division of focal plane (DoFP) polarization technology is to integrate polarization pixels on the sensors to detect polarized light in different directions, and the micropolarizing pixel and the detector pixel are aligned one by one [19]. Thus, all the polarization information of the target object can be obtained in a single frame. Compared with the traditional polarization detection methods, including division of time (DoT), division of amplitude (DoAm) and division of aperture (DoA), it has the advantages of compact structure, real-time imaging and high stability. With the development of micro-/nanoprocessing technology, the DoFP polarization detection technology has become a research hotspot [20,21,22,23,24,25].

Shortwave infrared imaging technology has the advantages of small atmospheric scattering, insensitivity to temperature, long effective detection distance and strong adaptability to complex weather conditions, so it has a wide range of application prospects in aerospace, reconnaissance, resource detection, archaeological identification, industrial detection, medical diagnosis and other fields. The indium gallium arsenide (InGaAs) detector has become the first choice for shortwave infrared detection systems because it can work at high operating temperatures and has excellent detectivity, uniformity, stability and good radiation resistance [26]. Due to their high sensitivity, high-speed detecting performance and good reliability, In_0.53_Ga_0.47_As/InP photodetectors are also promising for detecting shortwave infrared (SWIR) polarization. Through integration with a pixel-size grating, a single photosensitive pixel on the InGaAs focal plane array (FPA) would have the capability of detecting a certain polarization angle. Because of this, InGaAs FPA could capture an SWIR image with different polarization states in one shot, which effectively decreases the cost, the size and the power consumption of the polarization detecting system.

By using micro-/nanoprocessing technologies, including laser interference exposure, laser holographic exposure, electron beam lithography (EBL) and nanoimprint lithography (NIL), the development of micropolarizer array (MPA) suitable for the shortwave infrared band has made some progress [27,28]. The current DoFP shortwave infrared polarization detector usually adopts a periodically arranged superpixel structure, and each detection unit consists of 2 × 2 pixels to detect incident light in four different polarization directions. Due to the difficulty in achieving accurate matching between the polarization structures and photosensitive pixels as well as the crosstalk among the different polarization direction pixels, the currently reported key performance parameters such as the extinction ratio (ER) of the DoFP polarization detector cannot meet the needs of high-quality imaging.

In this paper, the InGaAs FPA detector integrated with a linear polarization grating (LPG) was proposed to solve the problem of mismatch and crosstalk. With the parameter design and optical simulation, the polarization InGaAs FPA detector was fabricated and systematically analyzed, based on our previous research on the DoFP InGaAs polarization detector [29]. By using back register lithography and EBL processes, the InGaAs detector on-chip integrated with LPGs was realized. By investigating three different widths of the polarization array, the structure of the nanograting array on the FPA was optimized and the measured ER was increased. Finally, the optical simulation of the nanograting-integrated InGaAs pixel was carried out to systematically discuss the mechanism of crosstalk and the limitations of the ER. Moreover, the impact of the grating geometry profile on the performance was analyzed, providing solutions for the further performance improvement of the polarization InGaAs detector.

## 2. Materials and Methods

The InP/In_0.53_Ga_0.47_As/InP material system with a double hetero-junction structure was fabricated on 350 μm InP (100) substrate by using the source molecular beam epitaxy (MBE) system. The elemental indium and gallium were adopted as group III sources. The As_2_ and P_2_ as the group V sources were cracked from arsenic and white phosphorus. The n-i-n structure for the fabrication of the PIN detector was composed of a 2 μm thick Si-doped InP contact layer with a concentration of 2 × 10^18^ cm^−3^, a 2.5 µm thick lightly doped In_0.53_Ga_0.47_As absorption layer with a concentration of 5 × 10^15^ cm^−3^, and a 1 μm thick n-type InP cap layer with a concentration of 1 × 10^16^ cm^−3^.

For the InGaAs FPA fabrication, a conventional planar structure device preparation process was adopted. The p-n junction of the device was buried in the absorption layer, which was beneficial to realize low dark current and noise because of the excellent surface passivation. A several-hundred-nanometers-thick layer of SiNx was firstly deposited on the epitaxy wafer via plasma-enhanced chemical vapor deposition (PECVD) and then was etched with the pattern of diffusion mask via inductively coupled plasma etching. To form a P-well of the photosensitive pixel, Zn atoms were diffused from the surface of the InP cap layer to the InP/InGaAs interface through mask apertures by means of the sealed-ampoule diffusion technique. Then, the p-electrode of the Au pattern was formed through the ion beam sputtering system and the lift-off process. The rapid thermal annealing process following the metallization was completed to realize the ohmic contact. To fabricate the n-electrode, the layers of InP and InGaAs around the photosensitive pixel arrays were etched to expose the n+-InP contact layer by combining wet etching and ion beam etching (IBE), followed by n-type ohmic metallization. The indium bumps were deposited on the p-electrode and n-electrode for the electrical interconnection. Finally, to realize the InGaAs FPA module, the as-prepared InGaAs photosensitive chip was flip-chip bonded with a silicon-based readout-integrated circuit (ROIC), which was used to provide appropriate bias for the InGaAs chip and output after reading the current signal generated by the chip.

In Figure 1 (left), the schematic diagram of a conventional backside-illuminated InGaAs FPA detector is given with the surface-integrated micro-/nanostructure. From the top down, it is the integrated polarization micro-/nanostructure and the InGaAs chip as well as the ROIC which are interconnected by indium bump coupling. The photo-generated carriers, after the incident light enters the InGaAs absorption layer, are collected by the p-n junction in the photosensitive pixel and then transferred to the ROIC through the indium bump to realize shortwave infrared imaging. From the perspective of optical parameter design, the InP substrate thickness should be optimized to suppress the signal crosstalk while the effective compatibility of the fabrication process should remain.

The subwavelength LPG integrated on the InGaAs FPA detector consists of four rows of aluminum gratings with angles of 0°, 45°, 90° and 135°, and each angle grating covers a row of photosensitive pixels (Figure 1, right). The structural parameters of the LPG are optimized by calculating the transmittance of the transverse magnetic (TM) waves and transverse electric (TE) waves with the finite difference time domain (FDTD) method, and the designed height and period are 100 nm and 300 nm, respectively, with a duty cycle of 0.5.

For the superpixel structure polarization detector, the area of each photosensitive pixel is larger than the matched grating cell in order to suppress crosstalk between different polarization angles, which resulted from the alignment deviation. As for the LPG-integrated detector, the different angle gratings are separated from each other, and the gratings completely cover the photosensitive region of the pixels to improve the responsivity.

Three groups of gratings of different widths were integrated on the same detector to evaluate the relationship between the polarization performance and the structural parameters, and each group of gratings covers 1024 × 4 pixels. Considering the relatively high ratio of the length to the width of the grating, the cumulation of the writing field stitching error during the EBL process may make the grating tilt or even cross rows. In order to ensure that the grating completely covered the corresponding photosensitive pixel with a center distance of 30 μm, the grating widths were designed as 40 μm, 50 μm and 60 μm, respectively, which meant the allowable position offsets of the three groups of gratings were 5 μm, 10 μm and 15 μm, respectively. The center distance between the adjacent two rows of gratings was set to 90 um, and the space between the gratings was shielded with an aluminum layer to prevent optical crosstalk. The arrangement of each group was 0°, 45°, 90° and 135° from bottom to top, and there was also metal shading in the area below the 0° grating and above the 135° grating.

For accurate matching of the grating to the pixel, the LPGs were prepared on the InGaAs chip via the fabrication process flow shown in Figure 2; before the chip was coupled to the ROIC, it was optimized on the basis of the conventional device process. The chip was first thinned to about 100 μm through the chemical mechanical polishing (CMP) process, thereby reducing the distance from the incident light modulated by the grating to the photosensitive pixel and improving the flatness. The inductively coupled plasma chemical vapor deposition (ICPCVD) process was then used to deposit a 100 nm thick SiO_2_ dielectric film on the polished chip substrate to control the influence of surface plasmonization on the transmittance and ER. The mask layers consist of 20 nm thick Ti and 100 nm thick Pt were deposited via the electron beam evaporation (EBE) process. Then, cross-shaped metal marks were realized with back register lithography (BRL) and the IBE processes, used for subsequent grating integration recognition, to achieve the pixel-level alignment of gratings and pixels. The aluminum layer with a thickness of about 100 nm was grown using the magnetron sputtering (MS) process on the surface of the detector. The designed grating pattern was transferred to the metal surface via the EBL process. In order to avoid the actual deviation from exceeding the design error allowable value, the exposure of the 1024 × 4 pixel LPG was completed with segmented alignment exposure. Through the reactive ion etching (RIE) process, the integration of the LPG on the chip was realized. Finally, the coupling interconnection of the chip and ROIC was achieved using flip-chip technology, and the preparation of the InGaAs polarization detector was accomplished (Figure 3).

The polarization performance of the detector at a specific wavelength was measured using the monochromator test system (Figure 4). The light source part was composed of a tungsten halogen lamp, a monochromator, a collimator and an adjustable aperture. The optical modulation section was composed of a linear polarizer, a quarter wave plate and a gran calcite polarizer to provide monochromatic illumination with polarization angles of 0−360° and wavelengths covering 0.9–1.7 μm. The FPA signal acquisition system was used to record the detector response signal at different polarization angles. With this system, the response spectrum of the detector could also be obtained by measuring its response signal at different wavelengths. Then, the spectrum was normalized with the tested signal of the standard InGaAs device under the same conditions and its actual responsivity.

For the polarization detector, the metal grating is aligned with the photosensitive pixel and the position relationship has been fixed, so the transmittance with different angles is calculated as follows:(1)$$T(\theta )=\frac{I_{g}}{I_{n}}$$
where *I_g_* is the signal of the photosensitive pixel covered by the metal grating, and *I_n_* is the response of the conventional pixel. The calculation formula for the *ER* of the metal grating is:(2)$$ER=\frac{T_{max}}{T_{min}}$$
where *T_max_* and *T_min_* are the maximum and the minimum values of the transmittance, respectively. 

## 3. Results

To investigate the effect of LPG width, the performances of gratings with three different widths were measured to carefully optimize the polarization performance under the wide spectrum. With the illumination of the halogen lamp spectrum through a monochromator, the FPA photosensitive pixels integrated with nanogratings of different polarization angles absorbed photons and generated the response current, which could be converted to the detected signal using the charge-coupling mode of ROIC. The variation of transmittance curves with different polarization angles, which tended to be a series of peaks and valleys, are shown in Figure 5. The ERs in Table 1 were calculated according to the transmittance. With the increase in grating width, the transmittance of the TM wave and the TE wave both increased to a certain extent. However, the value of the TE wave increased with a larger ratio, directly causing the deterioration of the ER for all of the gratings.

For wide-band polarization detection, the response signal and transmittance of the pixels covered with an LPG of 40 μm width were measured under three wavelengths: 1064 nm, 1310 nm and 1550 nm. The curves of the pixel signal and the optical transmittance with the variation of polarization angles are shown in Figure 6. At the wavelength of 1064 nm, the transmittance of TM waves was less than 70%, while at 1310 nm and 1550 nm, the transmittance of TM waves was above 90%, which was consistent with the simulation results. The average ER of the four angle gratings was 22:1, 29:1, and 46:1 at 1064 nm, 1310 nm and 1550 nm, respectively, and the peak ER of the detector reached 53:1 (Table 2), which was higher than that of the detector integrated with superpixel structure gratings [28]. The average ER of the four angle gratings could still reach 29:1 in the wide spectral range.

The normalized response spectrum of the 1024 × 4 integrated polarization InGaAs detector covered a 0.9–1.7 um band, as shown in Figure 7. The results showed that the spectrum of the polarization pixels with four angles was almost identical, implying that the fabrication process and actual structure parameter were quite reliable. Also, the response spectrum uniformity of the polarization detector will benefit applications such as infrared imaging and the hyper spectrum of a certain optical system.

## 4. Discussion

Compared with the nanogratings of superpixel structures, better polarization performance has been realized for the LPGs. Since the pixels with different polarization angles are optically separated on the LPGs, the crosstalk between adjacent pixels can be effectively suppressed, implying a less harmful impact on TM wave and TE wave propagation.

As previously mentioned in Figure 5, increasing the nanograting width, vertical to the direction of the InGaAs linear arrays, will significantly increase the transmittance of a single pixel. For the 60 μm width gratings, the pixel transmittance at some polarization angles is even higher than 1, which means the responsivity of the grating-integrated pixel is higher than the pixel without nanograting on the same InGaAs FPA. Since wider nanogratings could induce more severe crosstalk from the adjacent pixel, decreasing the grating width will effectively suppress crosstalk and increase the ER, as shown in Table 1.

For the FPA crosstalk investigation, the optical simulation of the grating-integrated InGaAs pixel was carried out (Figure 8a). The incident electric field is mainly distributed in the illuminated 30 μm pitch Pixel 1 from −15 to 15 μm in the X-direction. Due to the scattering properties of the metal nanograting and boundary, a weak electric field through the oblique propagation direction is observed in the unilluminated Pixels 2 and 3. Along with the propagation depth, the scattered electric field gradually widens, indicating that the crosstalk effect becomes serious. The extracted |E|^2^ curves with the setting optical propagation depth (black arrows in Figure 8a) are shown in Figure 8b. The |E|^2^ distribution in the pixels at 0.5 μm is almost the same as that of 0.01 μm, while the |E|^2^ distribution in the region of Pixel 2 and Pixel 3 significantly increases at a propagation depth above 10 μm.

For the as-prepared InGaAs FPA in this work, the thickness of the InP substrate was retained at about 100 μm to meet the needs of the fabrication process, such as manual operation and flip-chip bonding. Since the InGaAs FPA is a backside-illuminated device, the propagation distance of incident light should be 100 μm before arriving at the absorption layer. The curves of the calculated crosstalk rate are shown in Figure 8c. According to the simulation results, 40 μm width grating clearly outperforms the 50 μm and 60 μm wide grating, agreeing with the measured transmittance and ER in Figure 5 and Table 1. The limitation of the ER given in Figure 8d is based on the calculation of crosstalk rate and pixel fill factor. For the FPA integrated with a 40 μm width grating, the maximum theoretical value of the ER was 55 due to the optical crosstalk.

The as-prepared LPGs were characterized using a scanning electron microscope (SEM). Five points of each LPG of the three groups were selected and the linewidth was tested, and the average linewidth of 152.8 nm with a variation of less than 20 nm was achieved, indicating the excellent uniformity of the EBL and RIE processes. The geometry profile of the as-prepared nanograting, as shown in the SEM photo of Figure 9, deviates from the cross-section of a perfect rectangle which was used in the simulation work. The transmittance of the TM wave and ER of the micro-/nano-gratings with three different cross-sectional structures were calculated using the FDTD method, including rectangular and parabolic as well as the rectangular and parabolic stacking (Figure 10a), and the grating thickness (T) was set at 100 nm, respectively. The period (P) of the grating was set at 300 nm, and the width (W) was set at 150 nm. The results showed that, with the same grating thickness, the maximum and minimum ER of the grating could be explained by the deviation of the actual prepared grating morphology from the design structure. The cross-section of the aluminum grating after EBL processing was achieved with the rectangular and the parabolic cross-section, respectively. Therefore, the practical way to improve the ER of the detector is to optimize the grating preparation process, especially EBL and RIE, to enhance the structure’s accuracy.

## 5. Conclusions

In this paper, a 1024 × 4 shortwave infrared InGaAs FPA detector on-chip integrated with LPGs was carried out. In order to optimize the micro-/nanostructure, three groups of gratings with different widths were integrated on a single chip, and metal covers were used as a physical isolation between different angle gratings to prevent non-polarized light incidence. The polarization performance test results showed that, with the decrease in the grating width, the signal crosstalk was suppressed, and a higher ER was obtained. Finally, the pixels with LPG of 40 μm width displayed good polarization performance throughout the 0.9–1.7 um band, and the ERs at 1064 nm, 1310 nm and 1550 nm reached up to 22:1, 29:1 and 46:1, respectively, which were higher than the performance of the reported superpixel structure detector. The measured response spectrum of the polarization pixels with different angles was almost identical, which indicated the uniformity of the fabrication process and was beneficial for the imaging of the detection system. For the FPA crosstalk investigation, the optical simulation of the grating-integrated InGaAs pixel was carried out, and the limitation of ER is calculated. The result showed that the scattering of incident light in the InP substrate led to the crosstalk. Moreover, the deviation of the actual grating morphology from the designed structure caused a further reduction in the ER, and so the performance of the polarization detector could be further promoted by improving the grating preparation process and the thinning of the InP substrate. The results show that the shortwave infrared InGaAs detector on-chip integrated with subwavelength LPGs exhibits good performance, and further improvements are expected in the future by suppressing crosstalk, so it is an ideal choice for reducing the size, weight and power (SWaP) of infrared polarization imaging systems.

## Figures and Tables

**Figure 1 nanomaterials-13-02512-f001:**
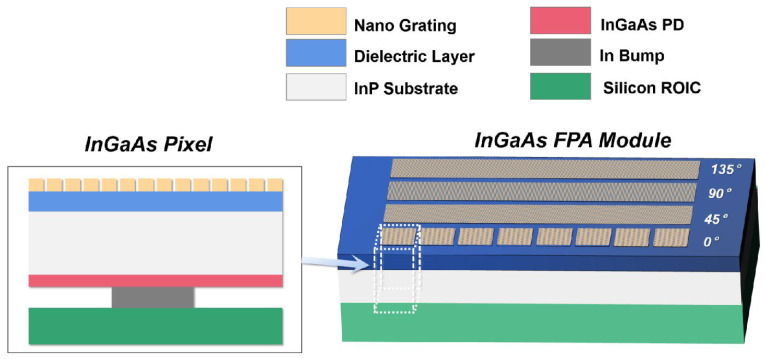
Structure diagram of the backside-illuminated InGaAs detector (**left**) and the InGaAs FPA integrated with LPGs (**right**).

**Figure 2 nanomaterials-13-02512-f002:**
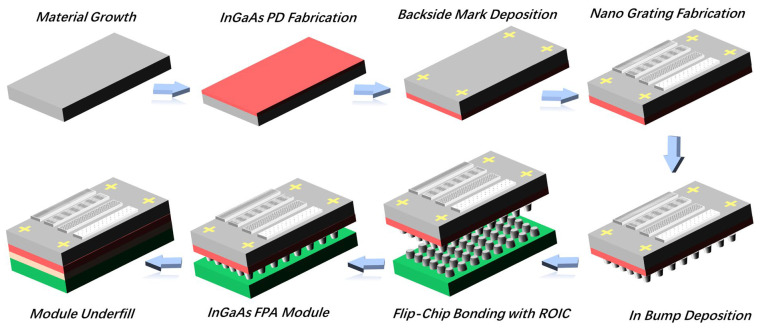
Schematic diagram of the fabrication process of gratings integrated on the InGaAs detector.

**Figure 3 nanomaterials-13-02512-f003:**
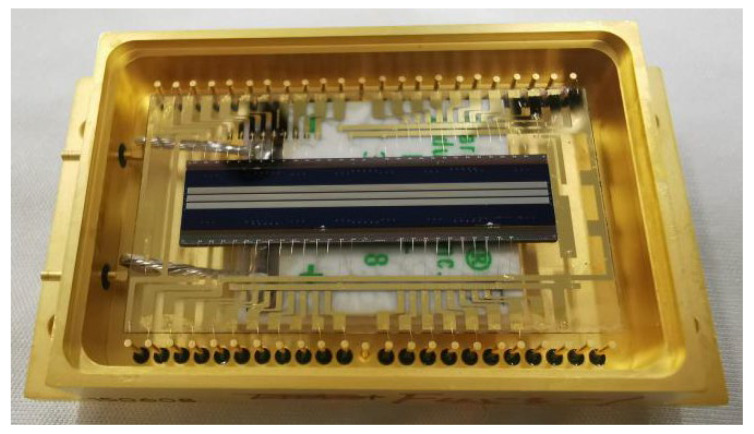
The fabricated InGaAs FPA detector integrated with LPGs.

**Figure 4 nanomaterials-13-02512-f004:**
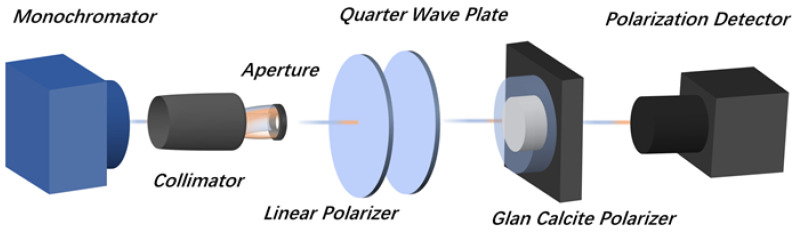
The polarization performance test system based on a monochromator.

**Figure 5 nanomaterials-13-02512-f005:**
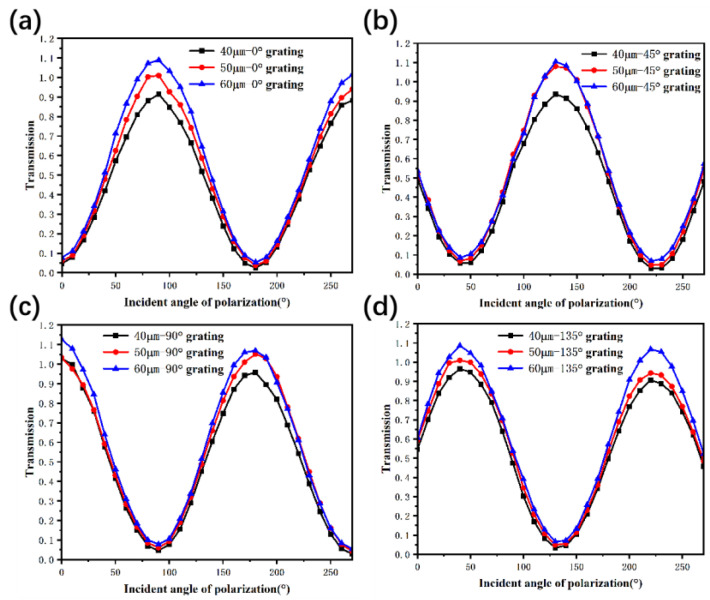
Transmittance of pixel gratings of (**a**) 0°, (**b**) 45°, (**c**) 90° and (**d**) 135° with different grating widths at incident angles.

**Figure 6 nanomaterials-13-02512-f006:**
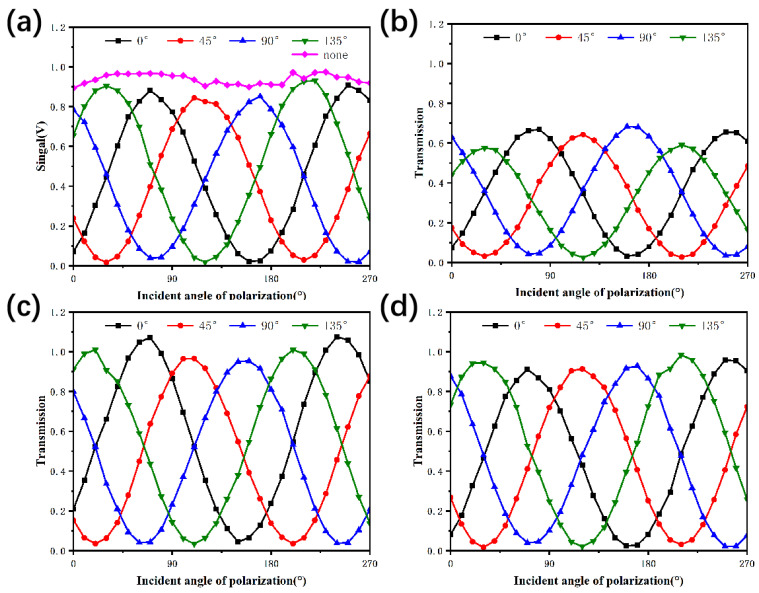
Polarization performance of the detector with LPG of 40 μm width: (**a**) response signals of pixels integrated with different angle gratings and conventional pixels at 1550 nm and transmittance of pixels integrated with different angle gratings at (**b**) 1064 nm, (**c**) 1310 nm and (**d**) 1550 nm.

**Figure 7 nanomaterials-13-02512-f007:**
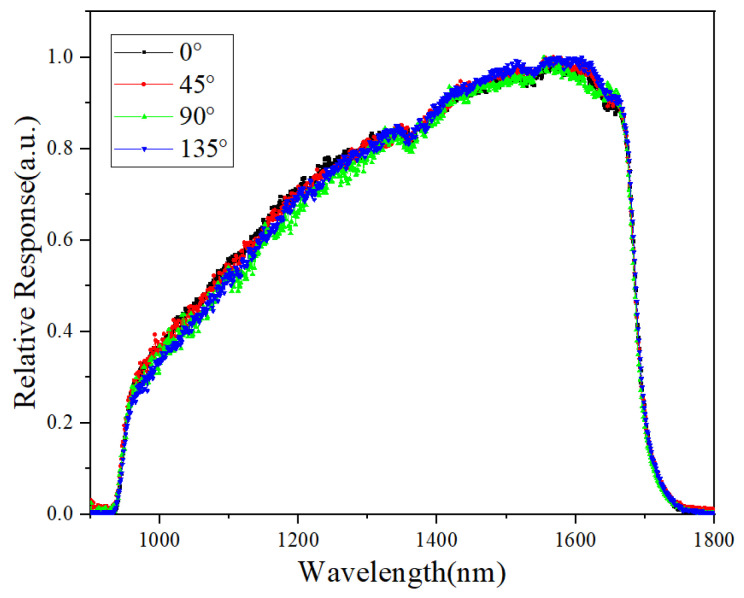
The response spectrum of the polarization pixels with four angles.

**Figure 8 nanomaterials-13-02512-f008:**
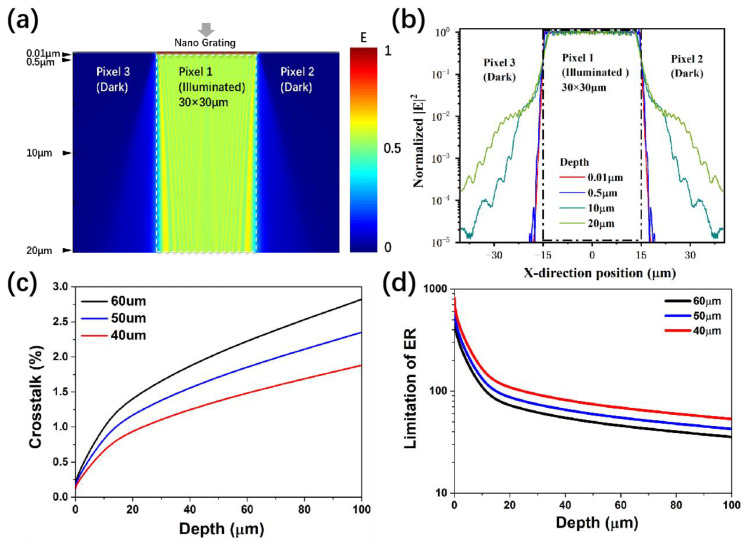
The optical simulation of the grating−integrated InGaAs pixel: (**a**) the electric field distribution of illuminated and dark pixels with 2D cross−sectional view, (**b**) the extracted |E|^2^ curves with different optical propagation depth, (**c**) the calculated crosstalk rate and (**d**) the limitation of ER line with different grating width.

**Figure 9 nanomaterials-13-02512-f009:**
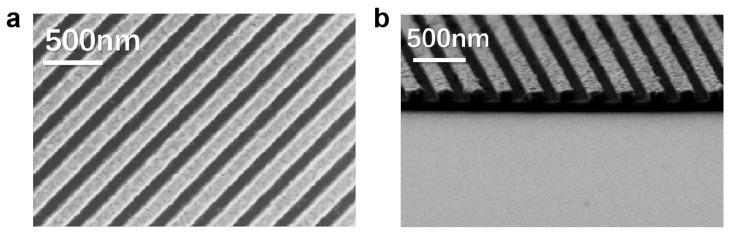
(**a**) Top view and (**b**) cross-section SEM photo of the fabricated Al nano-grating.

**Figure 10 nanomaterials-13-02512-f010:**
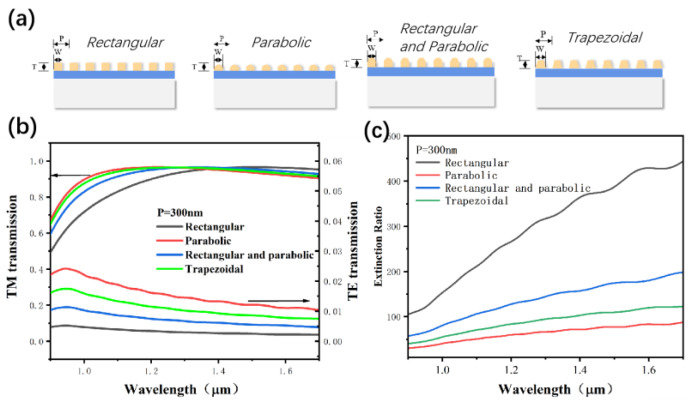
(**a**) The cross-sectional structure parameters for investigating the structure-dependent ER, (**b**) the simulated TM and TE transmittance and (**c**) the simulated ER of nano-gratings with the selected structure.

**Table 1 nanomaterials-13-02512-t001:** Measured ERs of the polarization InGaAs detector with LPGs of different widths.

Width of LPG	ER			
	0°	45°	90°	135°
40 μm	35:1	29:1	35:1	31:1
50 μm	27:1	21:1	25:1	23:1
60 μm	21:1	17:1	21:1	16:1

**Table 2 nanomaterials-13-02512-t002:** Comparison of measured ERs of the InGaAs detector with LPG of 40 μm width and the InGaAs detector with superpixel structure gratings at different wavelengths.

Wavelength	ER with LPG				ER with Superpixel Gratings [28]			
	0°	45°	90°	135°	0°	45°	90°	135°
1064 nm	21	24	20	24	15	18	17	18
1310 nm	24	29	28	36	12	11	9	16
1550 nm	40	53	43	48	9	12	8	8

## Data Availability

The data that support the findings of this study are available from the corresponding author upon reasonable request.

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
