# Peer review of "Shortwave Infrared InGaAs Detectors On-Chip Integrated with Subwavelength Polarization Gratings"

_nanomaterials, 2023, doi:10.3390/nano13182512_

Round 1

Reviewer 1 Report

In this manuscript, the authors present their successful development of a manufacturing technology for an imaging device based on InGaAs. The device demonstrates a remarkable high excitation ratio in the near-infrared spectrum. The practical applications of this technology in sensorics, biomedicine, and face recognition highlight its importance in the current context. 

Considering its potential impact, the manuscript is well-suited for publication in MDPI Nanomaterials without requiring further modifications.

Author Response

We appreciate the positive comments from the reviewer.

Reviewer 2 Report

 Huang and coworkers develop a 1024×4 shortwave infrared InGaAs  focal plane array detector on-chip integrated with  linear polarization gratings using several  groups of gratings on a single chip and resorting to metal covers to achieve isolation and prevent non-polarized light features, suppressing the signal crosstalk and enhancing the   extinction ratio. 

The experimental results are corroborated by  optical simulations.

Materials and Methods are well described and convincing and the results are clearly reported and robust.

Optical simulations, grating imaging and grating features dependance of TM and TE response are also reported and discussed.

The work seems quite novel and may be of interest for the community. 

this is fine

Author Response

(The authors gave the same response as above.)

Reviewer 3 Report

Ref_comments to the paper titled as “Shortwave infrared InGaAs detectors on-chip integrated with subwavelength polarization gratings” written by the authors:

Huijuan Huang, Xue Li, Yizhen Yu, Duo Sun, Guixue Zhang, Tao Li, Xiumei Shao and Bo Yang.

It is well known that different technical approaches and different materials the researchers are used in order to make the perspective detectors operated at the complex conditions. It is connected with the knowledge extending and with the useful evidence for the human life. Naturally, the problem of separating the image and the background, increasing the contrast of the implemented image is an urgent and technically important task. From this point of view the current article is actual and modern.

For the first, the authors have made the literature search, analyzing of 32 papers in the studied area. It is very good! Moreover, the authors have analyzed the manuscripts written on last 5 years. It is permit to tell that the authors have known the problem and can resolve the task set.

The paper is good illustrated. Materials and method section is interesting and permit to understand the device construction and testing. The formulas used are well known. In this concern, please estimate, apart from the estimation the extinction ratio (ER), the contract ratio as a ratio in transmission, when the magnitude of the difference in maximum and minimum is divided by the sum of the values of maximum and minimum transmission

Results section. It is nice data shown in Table 1. Measured ERs of the polarization InGaAs Detector with LPGs of different widths. Please make the comparative Table, when other polarization structures (with the features close to InGaAs materials can be done). It can extend your data with good advantage. Moreover, it is well known that materials based on InGaAs exhibit the luminescent properties. How was the photoluminescence kinetics calculated during the operation of your device? Furthermore, you have used the spectral range of 1064 nm, 1310 nm, and 1550 nm to estimate the ER in the conditions when the material widths and the angle for the testing have been varied. But, InGaAs materials can operate at the 1.7 micrometers as well. Have you the data about the ER in this spectral line?

Discussion part is good; it is coincided with our basic physical knowledge.

Conclusion should be extended, it is not accumulate the basic important data..

As for my local opinion, this paper can be published after the minor corrections.
